# Caterpillar-induced rice volatiles provide enemy-free space for the offspring of the brown planthopper

Xiaoyun Hu[1], Shuangli Su[1], Qingsong Liu[1,2], Yaoyu Jiao[1†], Yufa Peng[1], Yunhe Li[1]*, Ted CJ Turlings[3]

[1]State Key Laboratory for Biology of Plant Diseases and Insect Pests, Institute of Plant Protection, Chinese Academy of Agricultural Sciences, Beijing, China; [2]College of Life Sciences, Xinyang Normal University, Xinyang, China; [3]Laboratory of Fundamental and Applied Research in Chemical Ecology, University of Neuchâtel, Neuchâtel, Switzerland

**Abstract** Plants typically release large quantities of volatiles in response to herbivory by insects. This benefits the plants by, for instance, attracting the natural enemies of the herbivores. We show that the brown planthopper (BPH) has cleverly turned this around by exploiting herbivore-induced plant volatiles (HIPVs) that provide safe havens for its offspring. BPH females preferentially oviposit on rice plants already infested by the rice striped stem borer (SSB), which are avoided by the egg parasitoid *Anagrus nilaparvatae*, the most important natural enemy of BPH. Using synthetic versions of volatiles identified from plants infested by BPH and/or SSB, we demonstrate the role of HIPVs in these interactions. Moreover, greenhouse and field cage experiments confirm the adaptiveness of the BPH oviposition strategy, resulting in 80% lower parasitism rates of its eggs. Besides revealing a novel exploitation of HIPVs, these findings may lead to novel control strategies against an exceedingly important rice pest.

*For correspondence:
liyunhe@caas.cn

Present address: †Department of Entomology, University of Kentucky, Lexington, United States

Competing interests: The authors declare that no competing interests exist.

## Introduction

In their natural environment, plants interact with complex insect communities consisting of numerous species and different trophic levels (*Stam et al., 2014*; *Poelman and Dicke, 2014*; *Poelman, 2015*). Herbivore-induced plant volatiles (HIPVs) play key roles in these complex interactions (*Dicke and Baldwin, 2010*; *Schuman and Baldwin, 2018*; *Turlings and Erb, 2018*; *He et al., 2019*). For instance, HIPVs serve as cues for natural enemies, such as predators and parasitoids, to locate their prey or hosts (*Dicke et al., 2009*; *Allison and Daniel Hare, 2009*; *Allmann and Baldwin, 2010*; *Halitschke et al., 2008*; *Turlings and Erb, 2018*; *Schuman and Baldwin, 2018*; *Joo et al., 2018*). They also play a role in repelling herbivores that avoid inducible plant defenses and conspecific or heterospecific competition (*De Moraes et al., 2001*; *Knolhoff and Heckel, 2014*; *Anderson et al., 2011*; *Jiao et al., 2018*) or they can attract specialist herbivores that aggregate to collectively over-come the defense of their hosts (*Loughrin et al., 1995*; *Weed, 2010*; *Robert et al., 2012*). HIPVs can also be detected by neighboring plants and help them to anticipate an incoming attack (*Arimura et al., 2000*; *Heil and Ton, 2008*; *Engelberth et al., 2004*; *Karban et al., 2014*; *Sugimoto et al., 2014*; *Nagashima et al., 2018*). Hence, HIPVs provide information to all players in a plant's ecological network and through these various effects, HIPVs play a major role in determining the composition of insect communities in the field (*Xiao et al., 2012*; *Zhu et al., 2015*; *Poelman and Dicke, 2014*; *Blubaugh et al., 2018*; *Schuman and Baldwin, 2018*).

Here, we address the possibility that HIPVs induced by one species can be exploited by other herbivorous species to escape the attention of its natural enemies. Although there is evidence for

this from studies on invasive insects (*Shiojiri et al., 2002*; *Chabaane et al., 2015*; *Desurmont et al., 2018*), in all cases it remains unclear which and how volatile signals are involved in the interactions and if the observed behaviors enhance the herbivores' fitness. Here we studied this for the brown planthopper (BPH) *Nilaparvata lugens* (Stål). We had found that BPH exhibits a strong preference for rice plants (*Oryza sativa*) that are already infested by the rice striped stem borer (SSB), *Chilo suppressalis* (Walker) (*Wang et al., 2018*). BPH feeds on surface of rice plant stems, whereas SSB feeds inside the stems of rice plants (*Figure 1*). Based on the preference-performance relationship that is often observed for phytophagous insects (*Gripenberg et al., 2010*), we would expect that BPH perform better on SSB-infested plants than on uninfested plants. However, we only found a minor direct effect on BPH development if they feed on SSB-infested rice plants relative to uninfested plants (*Wang et al., 2018*). Therefore, we postulate that other ecological mechanisms provide a benefit that explains the observed preference of BPH for SSB-infested plants, one of which could be a reduced exposure to natural enemies.

Risk of predation or parasitism is an important factor influencing host selection behaviour of herbivores (*Denno et al., 1990*; *Holt and Lawton, 1993*), and we speculate that the BPH preference to share host plants with SSB may be linked to a reduced exposure to their natural enemies. The natural enemies of BPH are known to be attracted to BPH-induced rice volatiles (*Xiao et al., 2012*; *Lou et al., 2014*). We therefore hypothesize that the additional infestation by SSB affects the emission of HIPVs in a way that interferes with the attraction of natural enemies. Here we test this hypothesis for the most important natural enemy of BPH, the egg parasitoid *Anagrus nilaparvatae* (Pang et Wang) (*Figure 1*; *Lou et al., 2014*). We first conducted a series of oviposition preference tests to confirm that BPH females prefer SSB-infested to uninfested rice plants including wild rice. Next, we tested the parasitoid's behavioral responses to the odors of rice plants infested only by

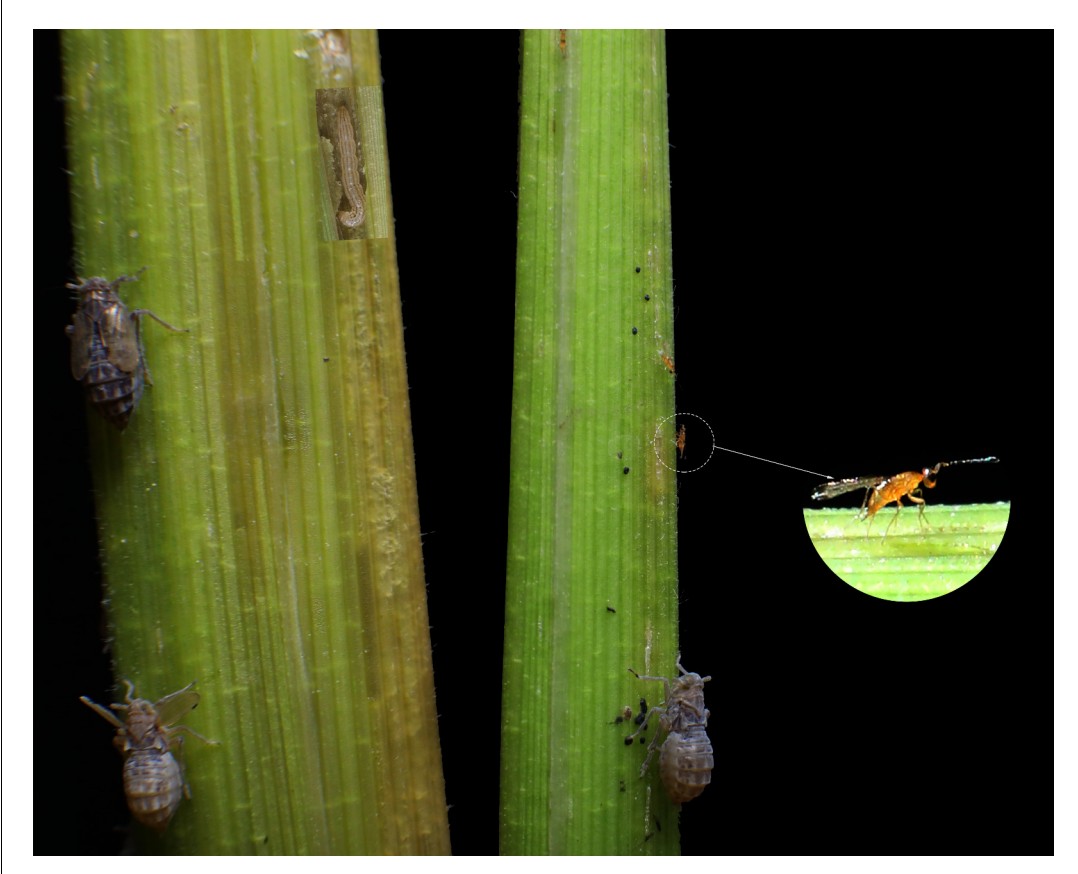

**Figure 1.** The study organisms on a rice plant, the brown planthopper *Nilaparvata lugens,* the rice-striped stem borer *Chilo suppressalis,* and the egg parasitoid *Anagrus nilaparvatae.*

SSB, only by BPH, or by both species. The role of specific HIPVs was then confirmed by first collecting, analyzing and identifying the volatiles released by SSB-infested, BPH infested and both species-infested rice plants and then measuring the behavioral responses of the parasitoid to synthetic blends of these volatiles. Finally, the actual fitness consequences of the BPH oviposition strategy were tested in greenhouse assays as well as field cages by determining the parasitization rates of BPH eggs by *A. nilaparvatae* on plants in the presence and absence of SSB larvae. The combined results provide conclusive evidence that the oviposition strategy of BPH females is adaptive and that heterospecific induction of plant volatiles provides enemy-free space for their progeny.

## Results

### Preferential settlement and oviposition of BPH on SSB-infested rice plants

When given a choice between uninfested rice and SSB-infested rice plants, the BPH females strongly preferred to settle on caterpillar-infested plants. This was the case for cultivated and for wild rice (Wilcoxon signed-rank test; all p<0.05) (*Figure 2A and B*). In concordance, BPH adult females laid significantly more eggs on caterpillar-infested rice plants than on uninfested plants (RT-test applied to a GLM, Piosson distribution error; all p<0.001) (*Figure 2A and B*).

### Odor preferences of *Anagrus nilaparvatae* wasps

We first examined the preferences of female *A. nilaparvatae* wasps when given a choice between the odor of infested plants (BPH alone, SSB alone, or both BPH and SSB) and the odor of uninfested plants (control). Whether caterpillars were present or not, increasing the BPH density positively correlated with the attraction of the wasps to infested plants (*Figure 3A*). In the absence of SSB, the parasitoid exhibited a preference for BPH-infested plants compared to insect free plants, although this was not significant at the lowest density of 2 BPH per plant (p=0.36, 0.002 and 0.001 at the density of 2, 5 and 10 per plant, respectively) (*Figure 3A*). When 1 SSB caterpillar was present, the parasitoids showed a strong preference for uninfested plants over infested plants either with 0 or 2 BPH per plant (p<0.001, in both cases), no significant preference with 5 BPH per plant (p=0.087), and significant preference for plants infested with 10 BPH over uninfested plants (p=0.001) (*Figure 3A*). With two caterpillars present, the parasitoids showed a strong preference for uninfested plants at densities of 0, 2 and 5 BPH per plant (p<0.01 in all three cases), and no preference at the density of 10 BPH per plant (p=0.46) (*Figure 3A*).

As expected, the parasitoids showed a strong preference for plants infested with 10 BPH over plants infested by 5 BPH (p=0.001) (*Figure 3B*). However, this preference was no longer observed when there was an SSB caterpillar present with the 10 BPH (p=0.87). Moreover, the parasitoids exhibited a significant preference for plants infested by 5 BPH vs. plants infested by 10 BPH together with two caterpillars (p<0.001) (*Figure 3B*).

### Rice volatile compounds affecting parasitoid behavior

A total of 41 compounds were detected in the headspace of uninfested rice plants, whereas 49 compounds were detected for rice plants damaged by BPH. The number of detected volatile compounds increased to 55 when the rice plants were infested by SSB only, or by both SSB and BPH. The relative quantities of specific compounds were significantly different among different plant treatments (*Figure 4—source data 1*). A projection to partial least squares-discriminant analysis (PLS-DA) using the contents of all detected volatiles showed a clear separation between herbivore-infested treatments and uninfested control plants, as well as between herbivore-treatments with or without SSB (*Figure 4*). The first two significant PLS components explained 30.8% and 10.3% of the total variance, respectively. The first component showed a clear separation between volatile profiles of plants with SSB infestation versus the other two treatments, while the second component separated volatile profiles released by plants infested by BPH only versus the other four treatments (*Figure 4*). However, the first two components could not separate the plants infested by SSB only or by SSB plus BPH. The volatile blends emitted by the plants of the three SSB treatments contained the same number and type of volatile compounds, and only a few of them showed significant difference in relative amount (*Figure 4—source data 1*).

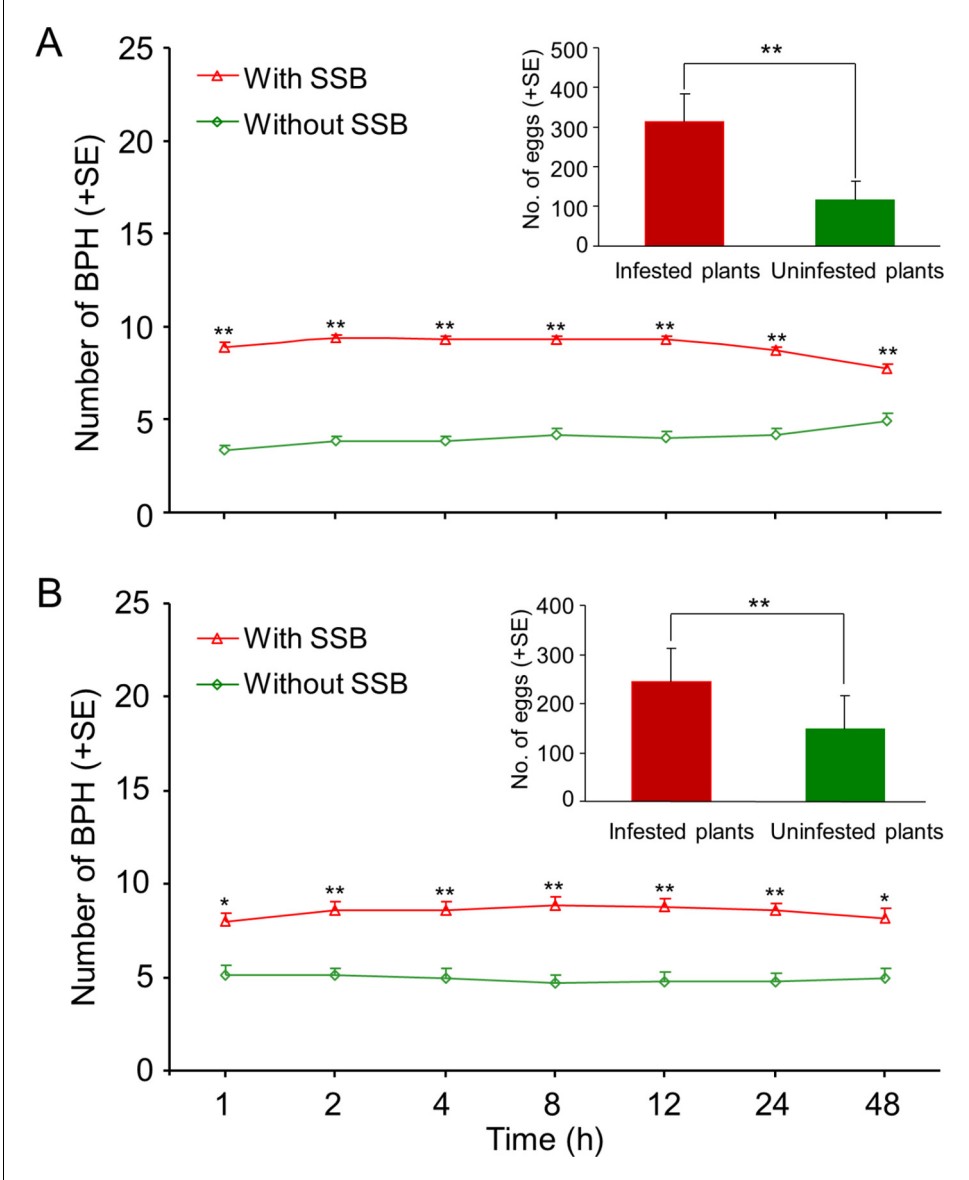

**Figure 2.** Planthoppers prefer to settle and lay eggs on caterpillar-infested rice plants. Preferences of BPH when given a choice between uninfested or SSB caterpillar-infested cultivated rice (**A**) and wild rice (**B**) plants were evaluated. Wild or cultivated rice plants were individually infested with two 3$^{rd}$ instar SSB caterpillars, and plants without caterpillar damage (uninfested plants) served as control. One day after the caterpillars had been placed on the plants, fifteen mated BPH females were released in the center of the cylindrical tube at equal distance from the two plants, and the number of BPH per plant was recorded for two consecutive days at different time points (1 hr, 2 hr, 4 hr, 8 hr, 12 hr, 24 hr and 48 hr) and the number of BPH eggs on the infested and healthy plants were counted at the end of the experiment. A Wilcoxon's signed-rank test for mean number of planthoppers and a likelihood ratio test (LR test) applied to a Generalized Linea Model (GLM, Poisson distribution error) for the mean number eggs. Asterisks indicate a significant difference within a choice test (*p<0.05, **p<0.01; N = 18–21).

The online version of this article includes the following source data for figure 2:

**Source data 1.** Number of BPH on caterpillar-infested or uninfested rice plants.
**Source data 2.** Number of BPH eggs laid on caterpillar-infested or uninfested rice plants.

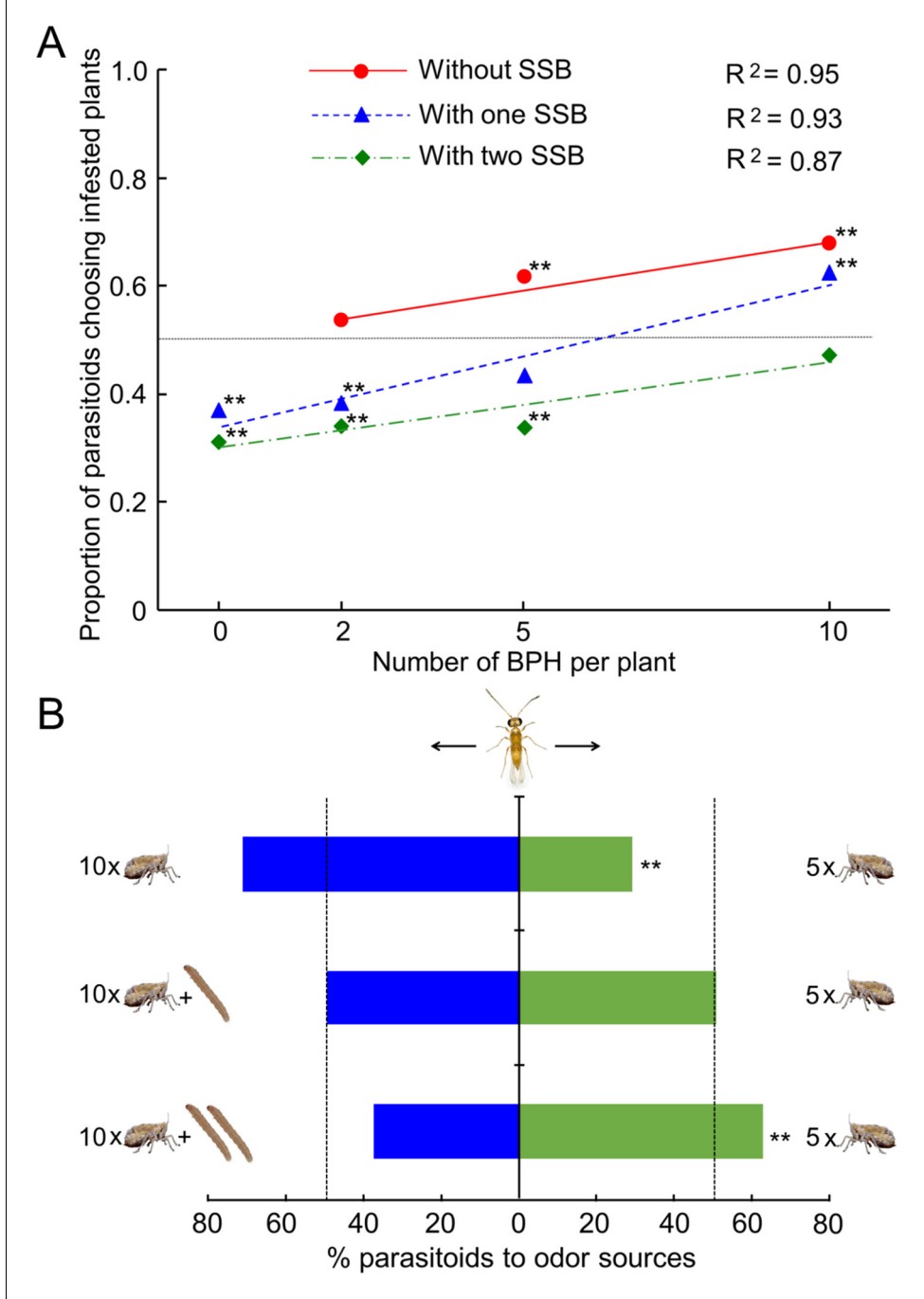

**Figure 3.** Choices of female *A. nilaparvatae* wasps in a Y-tube olfactometer. (**A**) Proportion of females choosing for the odor of BPH-infested plants without or with (one or two) SSB larvae, when offered next to the odor of control (insect-free) plants. (**B**) Percentage of females choosing for rice plants infested by 10 BPH without or with SSB caterpillars when offered next to the odor of rice plants only infested by 5 BPH. A LR test applied to a GLM (binomial distribution error) was used for the data, and the asterisks with the data points (**A**) or columns (**B**) indicate significant deviation from a 50:50 ratio (*p<0.05; **p<0.01; N = 89–102).

The online version of this article includes the following source data for figure 3:

**Source data 1.** Number of female *A. nilaparvatae* wasps choosing for the odors of differently treated plants.

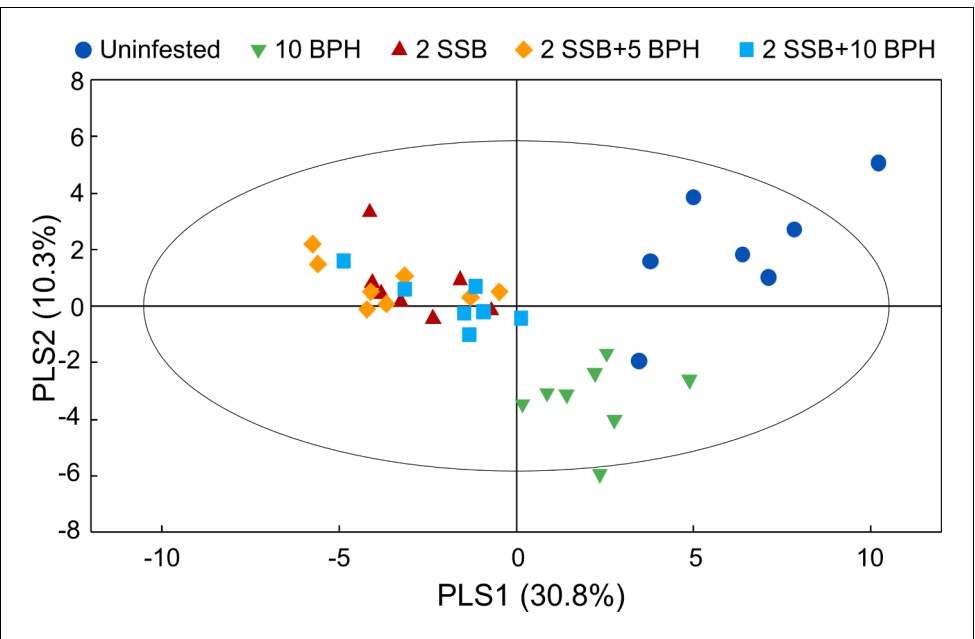

**Figure 4.** Partial least squares discriminant analysis (PLS-DA) of rice plant volatile compounds. The rice plants were either uninfested (Control), infested with 10 gravid BPH females for 12 hr (10 BPH), with two 3rd-instar SSB larvae for 12 hr then with 10 gravod BPH females for another 12 hr (2 SSB+10 BPH), with two SSB for 12 hr then with 5 BPH for another 12 hr (2 SSB+5 BPH), or with two SSB for 24 hr (2 SSB). The score plot display the grouping pattern according to the first two components and the ellipse defines the Hotelling's T2 confidence interval (95%) for the observations.

The online version of this article includes the following source data for figure 4:

**Source data 1.** Volatile compounds released by differently infested rice plants.

Based on the GC-MS results, we conducted Y-tube assays to assess the response of female *A. nilaparvatae* wasps to each of 20 volatile compounds that showed significant differences among treatments (*Figure 4—source data 1*). They were tested at a low and a high dose (*Figure 5*). Seven compounds, 2-nonanone, 2-tridecanone, (*E*)−2-heptany1 acetate, 2-heptanol, D-limonene, α-pinene and isopropyl myristate, had a significant repellent effect on the wasps compared to the control (hexane) at either low or high dose or both doses. In contrast, five compounds, DMNT, (*E*)-β-caryophyllene, linalool, methyl salicylate and (*E*)−2-hexenal, significantly attracted the wasps at both low and high doses. Interestingly, *A. nilaparvatae* females were attracted to TMTT at a low dose, but were repelled at a high dose (*Figure 5*). The remaining seven compounds had no effect on the wasps' behavior, even at a high dose; they were not included in the following experiments.

## Response of female *A. nilaparvatae* wasps to mixtures of volatiles

Next, we tested the response of female *A. nilaparvatae* to synthetic blends containing the 13 compounds that were found to affect the behavior of the parasitoid. There was a significantly higher percentage (73.1%) of *A. nilaparvatae* females that chose the synthetic blend that mimicked the volatiles emitted by plants infested by 10 BPH females compared to the control (only hexane) (p<0.001). By contrast, significantly fewer parasitoids chose the synthetic blends that mimicked the volatiles emitted by plants infested by 2 SSB caterpillars (30.4%; p<0.001) or two caterpillars plus 5 BPH females (37.3%; p=0.016). No significant effect on parasitoid preference was detected for the synthetic blend that mimicked the volatiles emitted by plants infested by 2 caterpillars plus 10 BPH females (p=0.33) (*Figure 6*).

## Parasitism rates of *N. lugens* eggs by *A. nilaparvatae* wasps

In the greenhouse experiment, gravid females of BPH were placed on rice plants contained in a plastic sleeve and therefore could not choose among plant treatments. In this no-choice situation, the

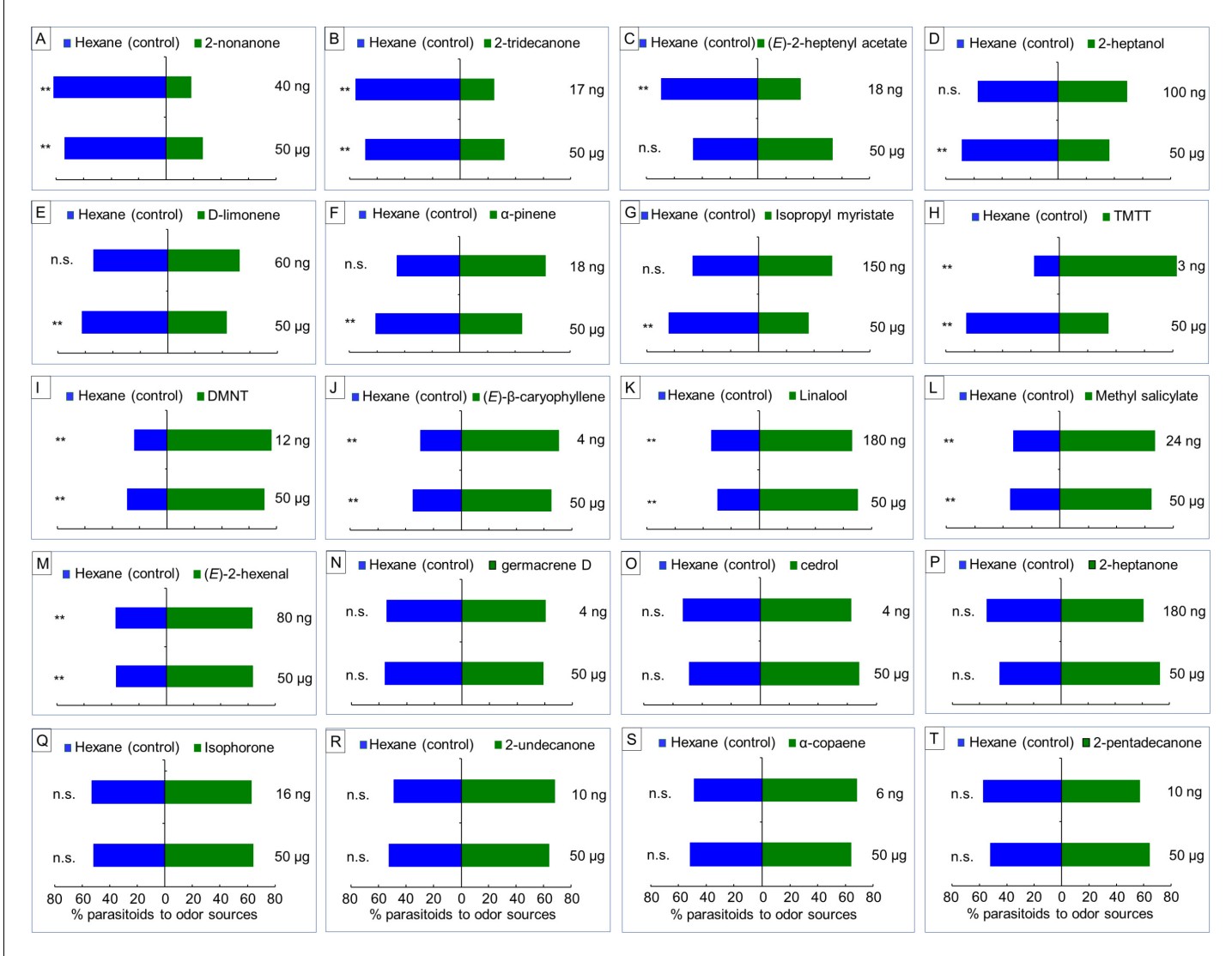

**Figure 5.** Parasitoid responses to individual herbivore-induced volatile compounds at low (3–180 ng) or high (50 μg) doses. Response of *A. nilaparvatae* (Y-tube assay) to selected key rice volatile compounds that were induced by damage of SSB, BPH or both herbivores. The compounds A-G exhibited were repellent at either a low or high dose or both doses, compound H was attractive at a low dose, but repellent at a high dose, whereas compounds I-M were attractive at both dosages. The remaining compounds (N-T) had no effect on the behavior of the parasitoid females at either dose. Columns with asterisks indicated the test volatiles significantly attract or repel the parasitoid (LR test applied to a GLM, binomial distribution error; *p<0.05, **p<0.01) (N = 50–122).

The online version of this article includes the following source data and figure supplement(s) for figure 5:

**Source data 1.** Responses of *A. nilaparvatae* wasps to individual synthetic volatile compounds.

**Figure supplement 1.** The emission pattern of 13 volatile compounds that affect parasitoid behavior.

**Figure supplement 1—source data 1.** Absolute concentrations of the 13 volatile compounds released by differently infested rice plants.

mean numbers of BPH eggs laid on rice plants were not significantly different between plant treatments (10 BPH vs. 10 BPH + 1 SSB, p=0.68; 10 BPH vs. 10 BPH + 2 SSB, p=0.69) (*Figure 7A*). Still, the rate of parasitism of BPH eggs was considerably lower on plants infested with 10 BPH plus 1 SSB or 2 SSB than on plants infested with BPH only (both p<0.01) (*Figure 7B*).

In the field cage experiment, gravid BPH females had a choice to select between differently treated rice plants for oviposition. As expected from the earlier results (*Figure 2*), they exhibited a significant oviposition preference for rice plants infested with both herbivores as compared to plants

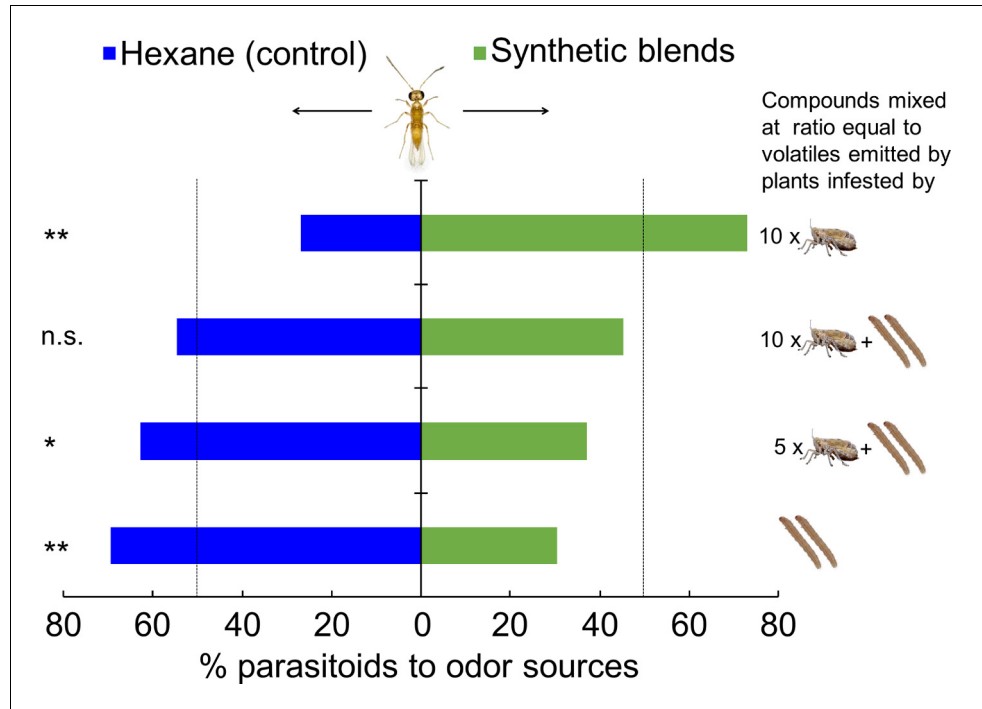

**Figure 6.** Response of *A. nilaparvatae* wasps to synthetic volatile blends. The synthetic blends contained volatile compounds at ratios equal to those detected in volatile blends emitted by rice plants that had been infested with 10 BPH only, 10 BPH plus 2 SSB larvae, 5 BPH plus 2 BPH larvae, or 2 SSB larvae only. Pure hexane was used as a control. Columns marked with asterisks indicate significant differences (LR test applied to a GLM, binomial distribution error; **p<0.01), and n.s. indicated a non-significant difference (p>0.05) (N = 60).

The online version of this article includes the following source data for figure 6:

**Source data 1.** Concentrations of 13 volatile compounds contained in each synthetic blend.

**Source data 2.** Choice of *A. nilaparvatae* wasps between pure hexane (control) and synthetic volatile blends.

infested with only BPH, with three times as many eggs on the doubly infested plants (p=0.004) (*Figure 8A*). Also, under these conditions, the BPH eggs were parasitized far less (over 5-fold reduction) on plants infested with both herbivore species than on plants infested with BPH only (p=0.005) (*Figure 8B*).

## Discussion

Many female insects have the formidable task to find suitable resources for their offspring. In the case of herbivores this involves localizing suitable host plants. These are not only plants that provide nutrition for optimal development, but also those with least risk that the offspring will be exposed to natural enemies (*Denno et al., 1990*; *Holt and Lawton, 1993*; *Feder et al., 1995*; *Shiojiri et al., 2001*; *Shiojiri et al., 2002*; *Ode, 2006*; *Joo et al., 2018*; *Brütting et al., 2018*). Here we reveal a clever strategy used by the BPH whereby it exploits the HIPVs induced by the SSB to find highly suitable, enemy-free rice plants.

Our first experiments confirm that the planthoppers exhibit a strong preference for SSB-infested plants whether it concerns cultivated or wild rice (*Figure 2*). This prompted the subsequent experiments to test the plant-mediated tritrophic interactions among SSB caterpillars, BPH planthoppers and its parasitoid *A. nilaparvatae*. The olfactometer assays showed the expected attraction of female *A. nilaparvatae* wasps to BPH-infested plants and that the attractiveness increased with an increase in the number of BPH per plant. However, when BPH-infested plants were co-infested by SSB caterpillars, wasp attraction to *A. nilaparvatae* was significantly reduced or even turned into repellency, depending on the densities of the respective insects on a rice plant (*Figure 3*). We tested the

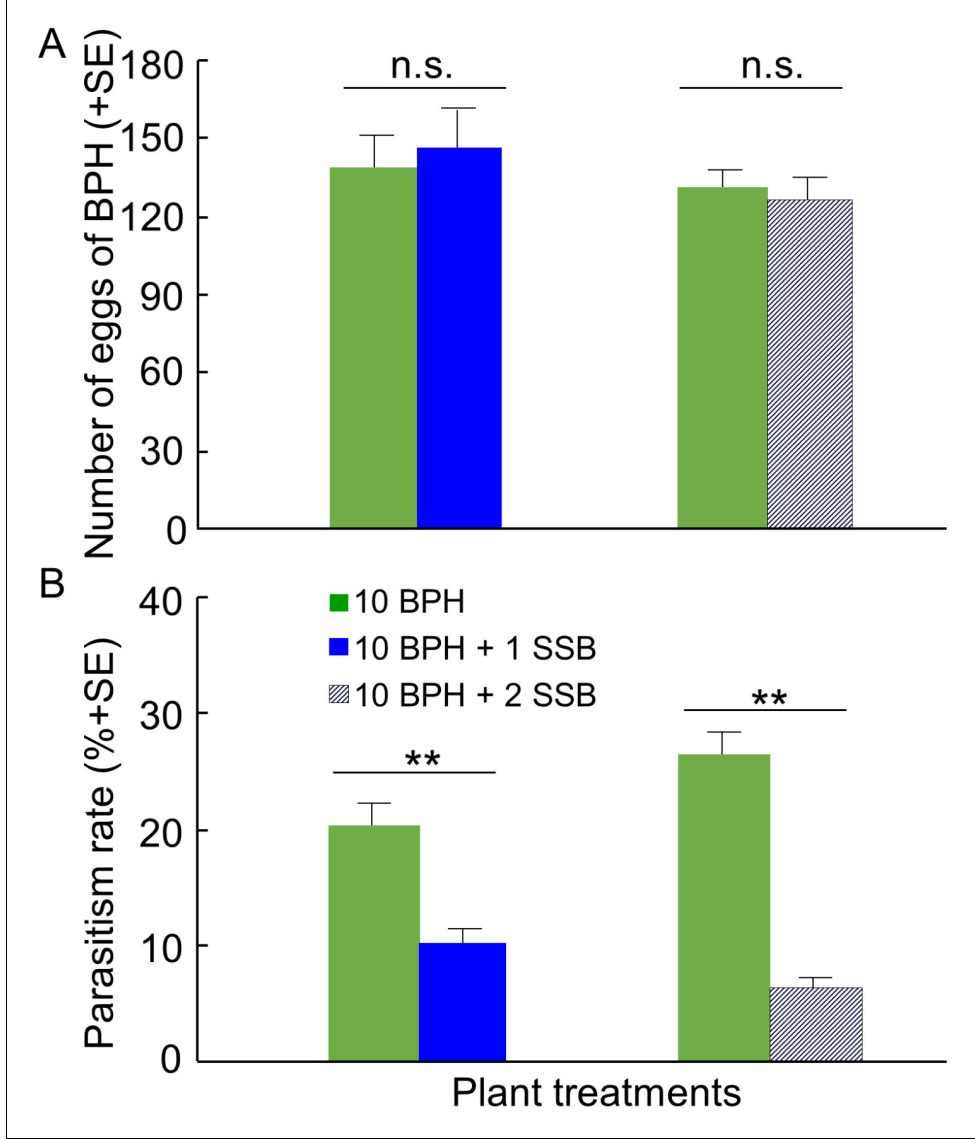

**Figure 7.** Parasitization rates of planthopper eggs by *A. nilaparvatae* in the greenhouse experiments. (**A**) Mean numbers of eggs deposited by BPH females on rice plants infested with BPH only or with both BPH and SSB, and (**B**) rates of parasitization of BPH eggs by *A. nilaparvatae* on the differently infested plants. LR tests applied to a GLM were conducted for the number of eggs (Poisson distribution error) and for the percentage of parasitized eggs (binomial distribution error). Columns marked with asterisks indicate significant differences (**$p < 0.01$) and n.s. indicated a non-significant difference ($p > 0.05$) (N = 14–15).

The online version of this article includes the following source data for figure 7:

**Source data 1.** Numbers of eggs deposited by BPH females and rates of parasitism of BPH eggs by *A. nilaparvatae* on the differently infested plants.

hypothesis that the distinct behavioral responses of the parasitoids to differentially treated rice plants are mediated by inducible volatiles, as it is known that HIPVs play a key role in the host-searching behavior of parasitoids (*Turlings and Wäckers, 2004*; *McCormick et al., 2012*; *Turlings and Erb, 2018*; *Joo et al., 2018*) and changes in the volatile blends may significantly reduce their attractiveness (*Chabaane et al., 2015*; *Desurmont et al., 2014*; *Desurmont et al., 2018*; *Blubaugh et al., 2018*).

Chemical analyses of the HIPVs emitted by rice plants in response to different herbivory showed considerable differences between treatments. PLS-DA of the volatiles revealed a clear separation between volatile profiles of plants with SSB infestation versus plants infested by BPH alone, but

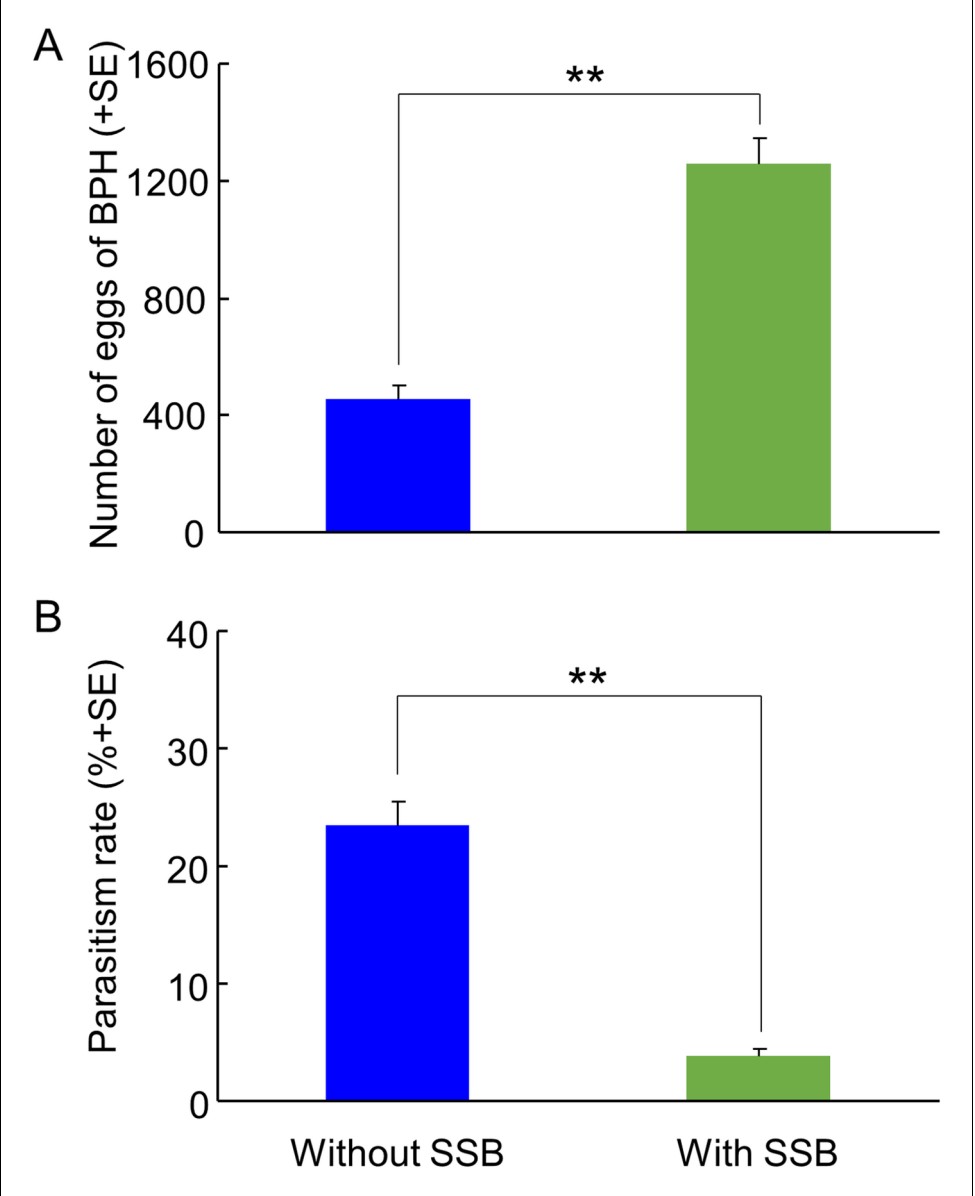

**Figure 8.** Parasitism rates of planthopper eggs by *A. nilaparvatae* in the field cage experiments. (**A**) Mean numbers of eggs deposited by BPH on rice plants that were infested with BPH only or with both BPH and SSB, and (**B**) rates of parasitism of BPH eggs by *A. nilaparvatae* on the differently infested plants. LR tests applied to a GLM were conducted for the number of eggs (Poisson distribution error) and for the percentage of parasitized eggs (binomial distribution error). Data columns marked with asterisks indicate significant differences (**p<0.01) (N = 5).

The online version of this article includes the following source data for figure 8:

**Source data 1.** Numbers of eggs deposited by BPH females and rates of parasitization of BPH eggs by *A. nilaparvatae* on the differently infested plants.

found no separation between plants infested by SSB only or by SSB plus BPH (*Figure 4*). This implies that SSB-induced volatiles dominate the plants' headspace and may thereby mask the BPH-induced volatiles. Based on the PLS-DA analyses, we expected that SSB caterpillars induce different volatiles than BPH and that this difference is responsible for the repellent effect on the parasitoid. Indeed, we identified (*E*)−2-heptenyl acetate as a compound that was only emitted by rice plants with SSB infestation, and it exhibited significant repellence to *A. nilaparvatae* (*Figure 5*). The parasitoid was also repelled by other volatiles, such as 2-nonanone, 2-tridecanone, 2-heptanol, D-limonene and α-

pinene that were found to be characteristic for SSB infestation. These compounds were also emitted by uninfested plants or plants infested by BPH only, but their concentrations were significantly increased upon SSB infestation (*Figure 5—figure supplement 1*). The results imply that, in combination with (*E*)−2-heptenyl acetate, these compounds are responsible for deterring *A. nilaparvatae* females when confronted with SSB-infested plants. It is expected that for a highly specialized parasitoid like *A. nilaparvatae* these responses are innate and not easily changed through experience (*Steidle and van Loon, 2003*; *Guo and Wang, 2019*), unlike in generalist parasitoids, which are notoriously good learners and adapt their responses to odor cues depending on positive or negative host location experiences (*Vet et al., 1995*).

We also considered the possibility that SSB infestation suppresses volatile compounds that are normally induced by BPH feeding. In other systems such suppression is known to reduce natural enemy attraction. For example, adding whiteflies to spider mite infested lima bean plants results in a reduction in the emission of (*E*)-$\beta$-ocimene, thereby reducing the plants' attractiveness to predatory mites (*Zhang et al., 2009*). Similarly, adding slugs to wild *Brassica rapa* that are infested by *Pieris brassicae* caterpillars results in an overall reduction in volatile emissions, leading to a reduced attraction of *Cotesia glomerata* (*Desurmont et al., 2016*). Yet, in the current study we did not find any evidence that the emission of parasitoid-attractive compounds was suppressed by SSB infestation. In fact, SSB infestation increased most of these attractants, just like the parasitoid-repellent compounds (*Figure 5*). This implies that changes in the overall composition and relative ratios of the volatile blend make plants that are co-infested by BPH and SSB unattractive to the parasitoid.

Although examples are still rare, the interference of one herbivore with the plant-mediated recruitment of natural enemies of another herbivore has been shown before, and new examples will certainly follow, especially in the context of climate change and the invasion of novel herbivores that come with it (*Desurmont et al., 2014*; *Desurmont et al., 2018*). For instance, when the exotic noctuid *Spodoptera littoralis* co-infests *Brassica rapa* plants that are under attack by native *Pieris brassicae* caterpillars, the dually-infested plants are considerably less attractive to the braconid wasp *Cotesia glomerata,* the main parasitoid of *P. brassicae* (*Chabaane et al., 2015*; *Desurmont et al., 2016*). Similarly, *Zhang et al., 2009* reported that when spider mite-infested lima bean plants are also infested by whitefly nymphs this results in a reduced attraction of predatory mites. As in our study, *Shiojiri et al., 2002* found that females of the moth *P. xylostella* preferentially oviposit on *P. rapae* -infested cabbage plants on which the parasitism rates of *P. xylostella* larvae by *C. plutellae* wasps are significantly reduced. Here we show that such an oviposition strategy can lead to tremendous fitness benefits.

Under greenhouse and semi-field condition, parasitism by *A. nilaparvatae* of planthopper eggs was considerably reduced on rice plants that were also infested with SSB larvae. In the greenhouse trial all plants carried equal numbers of BPH eggs (*Figure 7A*), In the field cage experiment, due to the strong preference of planthoppers for caterpillar-infested rice plants, more than twice as many eggs were laid on plants infested with both herbivores as compared to plants infested with planthoppers only. Nevertheless, the parasitism rate was more than 80% lower on plants infested with both herbivores. These results were well in accordance with the preferences that the parasitoid showed in the olfactometer experiments, implying that the lower parasitism rates on rice plants infested by both herbivores are caused by reduced attractiveness or even repellence by HIPVs released by SSB-infested rice plants. Our results imply that the presence of SSB larvae provides excellent protection for the offspring of planthoppers against parasitism by *A. nilaparvatae* and that the planthopper has evolved a highly adaptive oviposition strategy. Hence, rice plants with SSB infestation can be regarded as 'enemy-free/reduced space' for planthoppers (sensu *Denno et al., 1990*; *Holt and Lawton, 1993*; *Feder et al., 1995*; *Shiojiri et al., 2001*, *Shiojiri et al., 2002*; *Ode, 2006*).

*Stamp, 2001* and *Ode, 2006* postulate that herbivores can use plant secondary chemicals in defense against their own natural enemies, thereby creating enemy-free/reduced space. They mainly refer to insects that sequester plant defense compounds that provide resistance against the insects' natural enemies. In the known cases, the herbivores sacrifice their superior performance on optimal plants in order to obtain enemy-free/reduced space on less suitable plants (*Stamp, 2001*; *Ode, 2006*). For example, woolly bear caterpillars, *Grammia geneura*, not only feed on highly suitable and palatable *Malva parviflora* plants, but also on relatively toxic host plants, *Senecio longilobus* and *Ambrosia confertiflora,* containing pyrrolizidine alkaloids (*Singer et al., 2004*). Consumption of

the toxic plants decreases the performance of the caterpillar, but provides defense against parasitism by a common tachinid fly, *Exorista mella* (*Singer et al., 2004*). Such trade-offs result in dilemmas that many herbivorous species have to face (*Ode, 2006*). In sharp contrast, the oviposition strategy by planthoppers revealed in the current study does not involve any such trade-off. The 'enemy-free space' relies on plant volatiles induced by heterospecific species that have no apparent negative effect on the planthoppers. In fact, BPH performance may even be higher on SSB-infested rice plants, independent of parasitism risk. We know that SSB-infested rice plants contain higher amounts of nutritious amino acids, but the fitness of BPH on SSB-infested plants is only slightly higher than on uninfested plants (*Wang et al., 2018*). Hence, volatile-mediated 'enemy-free space' represents an 'optimized adaptive strategy' for herbivorous species (*Denno et al., 1990*; *Shiojiri et al., 2002*; *Knolhoff and Heckel, 2014*). Further insight into the exact chemistry responsible for attraction and repellency of BPH and its parasitoids may lead to the development of novel control strategies to mitigate the often devastating effects of BPH on rice yields (*Xiao et al., 2012*; *Lou et al., 2014*).

In brief, the current study demonstrates that HIPVs emitted by plants in response to feeding by a particular herbivore can be proactively utilized by another herbivore species to reduce the chances that their progeny fall victim to natural enemies. The oviposition strategy employed by the BPH is shown to be indeed adaptive, as it drastically reduces offspring mortality caused by an important egg parasitoid.

## Materials and methods

### Plants and insects

Seeds of cultivated rice (*O. sativa*), Minghui63 (MH63), and wild rice (*O. rufipogon*; erect type) were provided by Prof. Hongxia Hua (Huazhong Agricultural University, Wuhan, China), and Prof. Xinwu Pei (Institute of Biotechnology, Chinese Academy of Agricultural Sciences), respectively. Pre-germinated seeds were sown in the greenhouse at $27 \pm 3°C$ with $75 \pm 10\%$ RH and a photoperiod of 16:8 hr (light: dark). After 15 days, the seedlings were individually transplanted into bottom-pierced plastic pots (diameter 20 cm, height 18 cm) containing a 3:1 mixture of peat and vermiculite (Meihekou Factory, Meihekou, China). Potted plants were placed in a cement pool filled with 2 cm of water. The water was replaced weekly, and nitrogenous fertilizer was applied once per week before tillering and once every 2 weeks after tillering. Plants were used in the experiments 5 weeks after transplanting when they were at the tillering stage with 10–12 leaves on the main stem.

All three insect species, *C. suppressalis* (SSB), *N. lugens* (BPH) and *A. nilaparvatae* (*Figure 1*) were obtained from laboratory colonies. The colonies have been maintained in climatic chambers at $27 \pm 3°C$, $75 \pm 5\%$ RH, and 16: 8 h L:D for many generations with annual introductions of field-collected individuals. *C. suppressalis* larvae were reared on an artificial diet (*Han et al., 2012*) and 3rd instar larvae were used in the experiments. *N. lugens* were maintained on conventional rice plants, Taichung Native 1 (TN1) (*Wang et al., 2018*), and gravid females were used in the experiments. The parasitoid *A. nilaparvatae* was reared on *N. lugens* eggs. Adult wasps were fed 10% honey solution and maintained in glass tubes (diameter 3.5 cm, height 20 cm) for at least 6 hr to ensure mating, before females were used for the following experiments.

### Oviposition preference of BPH for uninfested and SSB-infested rice plants

To infest plants with SSB caterpillars, two 3rd instar caterpillars were starved for at least 3 hr and then placed on a wild or cultivated rice plant. The rice stems infested with caterpillars were covered with plastic sleeves to prevent insects from escaping (*Jiao et al., 2018*). Within 24 hr, the caterpillars drilled into the stems and caused visible damage to the plants. The caterpillars remained in the plants for the duration of the experiments. Uninfested plants were used as control.

For choice tests, intact whole rice plants were used. One day after the caterpillars had been placed on the plants, each plant was paired with an uninfested plant in H-tube olfactometers as described by *Wang et al., 2018*. Each olfactometer consists of a horizontally placed cylindrical plastic tube (diameter 8.0 cm, length 19.0 cm) with holes below and above on each side through which rice plants can be introduced. After gently inserting the main stems of the pair of plants on each end of the olfactometer, fifteen gravid BPH females were released in the center of the horizontal tube,

and the holes were plugged with sponges. After that, the numbers of BPHs that settled on each plant (SSB-infested or uninfested) were recorded for two consecutive days at different time points (1 hr, 2 hr, 4 hr, 8 hr, 12 hr, 24 hr, 48 hr). After the last time point, the BPH individuals were removed and the number of eggs laid on each rice plant was counted. This choice test was replicated 18–21 times.

### Response of *A. nilaparvatae* wasps to insect-infested rice plants

Multiple types of infested rice plants were prepared: SSB-infested, BPH-infested, and plants infested with both species.

i.   SSB-infested plants. Each potted rice plant was artificially infested with 1 or two third-instar SSB larvae that had been starved for >3 hr using the method as described above. After 24 hr feeding, rice plants were used for experiments, during which the caterpillars remained in the plants for the duration of all experiments, since they had bored into rice stems and it was not possible to get them off without artificial damage on plants. The wormholes were sealed using parafilm to avoid any volatiles released from insects.

ii.  BPH-infested plants. Each potted rice plant was infested with 2, 5 or 10 individuals of gravid BPH females. The planthoppers were removed from rice plants after 12 hr of feeding before the plants were used.

iii. Plant infested with both SSB and BPH. Rice plants were infested with 1 or two third-instar SSB larvae only as described above for the first 12 hr, then 2, 5 or 10 individuals of gravid BPH were additionally introduced on to each plant. After 12 hr of feeding, all planthoppers were removed and the caterpillars were kept in the plants, but the wormholes were sealed using parafilm before the plants were used in the olfactory behavior bioassays.

Dual-choice (Y-tube) olfactometers (10 cm stem; 10 cm arms at 75° angle; 1.5 cm internal diameter) were used to investigate the behavioral responses of gravid *A. nilaparvatae* females to rice plants that had never been infested by any insects (control plants) and each type of insect-infested rice plants as described above (treated plants). In addition, we further compared the preference of *A. nilaparvatae* females to rice plants that had been infested with 10 planthoppers plus 0, 1 or two caterpillars, and rice plants infested with five planthoppers only. Ten rice plants were enclosed in a glass bottle used as one odor source. Eight-twelve parasitic wasps were tested for each pair of plants (odor sources), and ten pairs of plants were used in each choice test (treatment), resulting in a total of 80–120 females tested for each treatment. The olfactory behavior assays were conducted using the method as described by *Liu et al. (2015)*. Individuals that made no choice within 5 min were excluded from the analyses. All tests were conducted between 10:00 and 16:00 in a climate-controlled laboratory room (27 ± 3°C, 40% RH).

### Collection and analysis of rice plant volatiles

Five types of rice plants were prepared: (a) plants that remained uninfested; (b) plants infested with 10 gravid BPH females for 12 hr; (c) plants infested with two 3rd-instar SSB larvae for 12 hr before additional introduction of 10 gravid BPH females for extended feeding of 12 hr; (d) plants infested with two 3rd-instar SSB larvae for 12 hr before additional introduction of 5 gravid BPH females for extended feeding of 12 hr; (e) plants infested with two 3rd- instar larvae of SSB for 24 hr. Before the plants were used for volatile collections, planthoppers were removed, and the caterpillars remained in the rice stems, but the wormholes were sealed using parafilm to avoid any volatiles released from the caterpillar themselves.

Volatiles emitted by rice plants were collected using a dynamic headspace collection system as described by *Jiao et al., 2018*. Ten plants were transferred into a glass bottle (3142 ml). Air was filtered through activated charcoal, molecular sieves (5 Å, beads, 8–12 mesh, Sigma-Aldrich), and silica gel Rubin (cobalt-free drying agent, Sigma-Aldrich) before entering the glass bottles. The system was purged for 30 min at 400 ml/min with purified air before attaching a tube filled with 30 mg Super Q traps (80/100 mesh, ANPEL Laboratory Technologies (Shanghai) lnc, China) to the air outlet in the lid to trap the headspace volatiles (*Jiao et al., 2018*). Volatile collection lasted for 4 hr (11:00-15:00) in a climate chamber at 27 ± 3°C, 75 ± 10% RH.

Volatiles were analyzed by gas chromatography coupled with a mass spectrometry system (Shimadzu GCMS-QP 2010SE using an RTX-5 MS fused silica capillary column). Samples were injected in a 1 μl volume with a splitless injector held at 230°C. The GC-MS was operated in the scan mode with

a mass range of 33–300 amu and was in an electron-impact ionization (EI) mode at 70 eV. The oven temperature was maintained at 40℃ for 2 min, and was then increased to 250℃ at 6 ℃ min$^{-1}$, where it was held for 2 min. Volatile compounds were identified by mass spectral matches to library spectra as well as by matching observed retention time with that of available authentic standards. If standards were unavailable, tentative identifications were made based on referenced mass-spectra available from NIST (Scientific Instrument Services, Inc, Ringoes, NJ, USA) or based on a previous study (*Xiao et al., 2012*; *Jiao et al., 2018*). Relative quantifications of compounds were first based on their integrated areas related to the internal standard (*De Lange et al., 2020*). This does not allow for very precise quantification, but provides accurate information for comparisons among volatile collections. We also use these values as rough estimates of quantity to select the dosages of chemical standards that we used in the first olfactometer assay.

## Identification of key rice volatiles that attract or repel *A. nilaparvatae*

Based on the GC-MS results, twenty volatile compounds including α-pinene, D-limonene, linalool, 2-heptanone, 2-heptanol, cedrol, (*E*)-β-caryophyllene, (*E*)−2-hexenal, 2-nonanone, isophorone, methyl salicylate, 2-tridecanone, isopropyl myristate, (*E*)−2-heptanyl acetate, α-copaene, TMTT, DMNT, germacrene D, 2-pentadecanone, and 2-undecanone were selected for further experiments to determine their effects on *A. nilaparvatae* behavior. The compounds were selected following the criteria: i) significantly increased or newly produced in response to damage by SSB or BPH or both and ii) having been reported to attract or repel *A. nilaparvatae* females in previous studies (*Xiao et al., 2012*; *Wang and Lou, 2013*). The compounds were purchased from different companies with analytical grade purity: 2-nonanone from Shanghai Aladdin Bio-Chem Technology Co., Ltd (Shanghai, China); DMNT from Enamine Ltd (Kiev, Ukraine); 2-tridecanone from Dr. Ehrenstorfer GmbH (Augsburg, Germany); TMTT, α-copaene and germacrene D from Toronto Research Chemicals Inc (Ontario, Canada); isophorone, (*E*)−2-heptanyl acetate, 2-pentadecanone, 2-undecanone, isopropyl myristate and (*E*)−2-hexenal from Tokyo Chemical Industry Co., Ltd (Shanghai, China), and the others from Sigma-Aldrich (St Louis, MO, USA).

Dual-choice (Y-tube) olfactometers (10 cm stem; 10 cm arms at 75˚ angle; 1.5 cm internal diameter) were used to investigate the behavioral responses of gravid *A. nilaparvatae* females to the selected compounds. The synthetic volatile compounds were individually dissolved in pure hexane, and each compound was tested at a high dose of 50 μg per 10 μl pure hexane and a low dose that was equal to its share in the volatile blend emitted by rice plant infested by SSB only (*Figure 5*). Filter papers (1 × 2 cm) were loaded with either 10 μl of the volatile solution or 10 μl pure hexane (control) and were, respectively, put into two glass jars (diameter 10.5 cm; height 10 cm) as a pair of odor sources. The procedure for the Y-tube assays was the same as described above. Fifty to 122 insects were tested for each compound.

## Response of female *A. nilaparvatae* wasps to mixtures of volatiles

The results from Y-tube assays revealed that 13 volatile compounds exhibited attraction or repellence to *A. nilaparvatae* females, either at high or low concentrations (*Figure 5*). To test representative blends of these compounds required absolute quantifications of these 13 compounds. For that, using synthetic samples, the response factor (R) of each individual compound relative to the internal standard was calculated using the equation: $R = (A_X/A_{IS})/(C_X/C_{IS})$. Here $A_X$ is the chromatography peak area and $C_X$ is the concentration of the analyte, whereas $A_{IS}$ is the chromatography peak area and $C_{IS}$ is concentration of the internal standard. With the equation, the response factor can be calculated from a calibration plot of $A_X/A_{IS}$ vs $C_X/C_{IS}$, whereby the response factor is the slope and the y-intercept is assumed to be 0 (*Figure 5—figure supplement 1—source data 1*). Thus, with the response factor, the absolute concentration of each tested volatile compound was calculated based on the ratio of a compound's peak area to the internal standard's peak area in each sample (*Kalambet and Kozmin, 2018*; *JoVE Science Education Database, 2020*). Using these values, synthetic blends containing the 13 compounds (11 compounds in the blend for the treatment 'BPH only') were prepared in concentrations that corresponded to the ratios of compounds detected in the collection of volatiles from rice plants infested with BPH only, SSB only or both (*Figure 6—source data 1*). Y-tube assays, as described above, were conducted in order to verify whether the

combinations of these volatiles were indeed responsible for the parasitoid's responses to the differently treated rice plants.

## Parasitism rates of *N. lugens* eggs by *A. nilaparvatae* wasps

### Green house experiment

Parasitism rates by *A. nilaparvatae* of BPH eggs were determined for plants infested with BPH only or plants infested with both BPH and SSB; three types of rice plants were prepared: i) rice plants were infested with 10 gravid females of BPH for 12 hr; ii) rice plants were infested with 1 SSB caterpillar for 12 hr before 10 gravid females of BPH were subsequently introduced to the plants for an additional 12 hr; iii) rice plants were infested with 2 SSB caterpillars for 12 hr before 10 gravid females of BPH were subsequently introduced to the plants for an additional 12 hr. Afterwards, all the planthoppers were removed from the rice plants and the caterpillars were left in the plants as described above. Subsequently the main stems with BPH eggs of a pair of rice plants were contained in a cylindrical plastic tube (*Wang et al., 2018*) and 5 pairs of newly emerged parasitic wasps (<1 day old) were released into each tube. After 72 hr, the rice plants with planthopper eggs were collected, and the total number of BPH eggs on each plant was counted and their parasitization status was determined under a microscope two days later. Each choice test was replicated 14 to 15 times. The experiment was performed in a greenhouse at 27 ± 3°C and with 75 ± 10% RH and a photoperiod of 16 L: 8 D.

### Field cage experiment

To further investigate the parasitism rate by *A. nilaparvatae* of BPH eggs on plants with or without SSB under a more realistic condition, a cage experiment was carried out in a field near Langfang City (39.5°N, 116.4°E), China. Rice seedlings (MH63) were obtained as described above, and three plants were transplanted into each clay pot, stimulating the common rice-cropping practice of three plants per hill in the fields. When the potted plants were at the end of the tillering stage with 14–16 leaves on the main stem, four pots were randomly selected and positioned in the four corners of a cage (42 cm length ×42 cm width ×70 cm height) made of 80-mesh nylon nets. In each cage, plants in two pots on a diagonal line were infested with 3$^{rd}$ instar SSB larvae (one insect per tiller); the other two plants remained intact. After 24 hr, fifty gravid BPH females were additionally released into each cage by placing an open plastic box containing the insects in the center of the cage. Next, after 12 h of BPH ovipostion, 20 pairs of freshly emerged parasitoids were released into each cage. After an additinal 72 hr, the rice plants with planthopper eggs were collected, and the total number of BPH eggs on each plant was counted and their parasitization status was determined under a microscope two days later. The experiment was repeated five times.

## Statistical analyses

All data were checked for normality and equality of variances prior to statistical analysis. Wilcoxon's signed-ranks tests were used to compare mean number of planthoppers settled on rice plants, while a likelihood ratio test (LR test) applied to a Generalized Linea Model (GLM) were conducted to compare the mean number of planthopper eggs on rice plants (Poisson distribution error) and parasitism rates of BPH eggs by *A. nilaparvatae* (binomial distribution error). Behavioral responses of *A. nilaparvatae* in Y-tube assays were analyzed with a LR test applied to a GLM (binomial distribution error), with an expected response of 50% for either olfactometer arm. For analyses of volatiles collected from differently treated rice plants, one-way ANOVAs were conducted after the data were fourth-root transformed. Differences for specific compound between treatments were determined using the Tukey HSD test. The data on volatile emissions were further investigated by discriminant analysis. The data were normalized by medians, then auto-scaled (mean centered and divided by the standard deviation of each variable) before being analyzed using partial least squares-discriminant analysis (PLS-DA). All statistical analyses were conducted with SPSS 22.0 (IBM SPSS, Somers, NY, USA), except for the PLS-DA were performed using SIMCA 14.1 software (Umetrics, Umeå, Sweden).

## Acknowledgements

We thank Prof. Hongxia Hua (Huazhong Agricultural University, Wuhan, China) and Prof. Xinwu Pei (Institute of Biotechnology, Chinese Academy of Agricultural Sciences) for kind provision of rice seeds. We also thank Dr. Changlong Shu (Institute of Plant Protection, CAAS, Beijing, China) for his advice on graphic production. The study was supported by the National Natural Science Foundation of China (31972984) and the National GMO New Variety Breeding Program of PRC (2016Z × 08011–001). The contribution by TCJT was supported by European Research Council Advanced Grant 788949.

## Additional information

### Funding

| Funder | Grant reference number | Author |
| --- | --- | --- |
| National Natural Science Foundation of China | 31972984 | Yunhe Li |
| National GMO New Variety Breeding Program of PRC | 2016ZX08011-001 | Yufa Peng |
| H2020 European Research Council | Advanced Grant 788949 | Ted CJ Turlings |

The funders had no role in study design, data collection and interpretation, or the decision to submit the work for publication.

### Author contributions

Xiaoyun Hu, Formal analysis, Investigation, Writing - review and editing, Study design; Shuangli Su, Investigation; Qingsong Liu, Formal analysis, Writing - review and editing, study design; Yaoyu Jiao, Writing - review and editing; Yufa Peng, Funding acquisition, Writing - review and editing; Yunhe Li, Conceptualization, Formal analysis, Supervision, Funding acquisition, Methodology, Writing - original draft, Writing - review and editing, study design; Ted CJ Turlings, Formal analysis, Methodology, Writing - review and editing

### Author ORCIDs

Yunhe Li (iD) https://orcid.org/0000-0003-0780-3327
Ted CJ Turlings (iD) https://orcid.org/0000-0002-8315-785X

### Decision letter and Author response

Decision letter https://doi.org/10.7554/eLife.55421.sa1
Author response https://doi.org/10.7554/eLife.55421.sa2

## Additional files

### Supplementary files

- Transparent reporting form

### Data availability

Source data have been provided for data in all figures as additional data files.

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
