## [Decision Letter]

**Acceptance summary:**

We find the study to be elegantly performed and well written. It tests an interesting and relatively new ecological hypothesis and investigates both function and mechanism, in both controlled and realistic environments. This is unusually comprehensive. The study system (rice, pests, biocontrol agents) is of broad general interest.

**Decision letter after peer review:**

Thank you for submitting your article "Caterpillar-induced rice volatiles provide enemy-free space for the offspring of the brown planthopper" for consideration by *eLife*. Your article has been reviewed by three peer reviewers, including Meredith C Schuman as the Reviewing Editor and Reviewer #1, and the evaluation has been overseen by Ian Baldwin as the Senior Editor.

The reviewers have discussed the reviews with one another and the Reviewing Editor has drafted this decision to help you prepare a revised submission.

Summary:

We find this study to be elegantly performed and well written. It tests an interesting and relatively new hypothesis regarding insect oviposition choices, biocontrol by parasitoids, and enemy-free space for herbivores in an economically important system. The authors include data from an important agricultural plant and a wild relative, and investigates both function and mechanism, in both controlled and realistic environments.

However, we have several concerns regarding missing information, and methods, which must be resolved before we can determine whether the study is appropriate for publication in *eLife*.

Therefore we invite the authors to submit a revised version of their manuscript which should fully address the following concerns.

Essential revisions:

1) The relative quantification of at least those volatiles used for bioassays needs to be done rigorously, e.g. using response factors, and the amounts of pure compounds tested in bioassays need to be compared to amounts calculated to be present in plant blends, at least based on response factors. All three reviewers agree that this is a critical point which must be resolved. Relative quantification in terms of percentage internal standard only does not support any conclusions about absolute amounts of volatiles (ng-ug emitted per plant per time) or their ratio, and thus cannot be used as a basis to select concentrations for bioassays or to interpret bioassay results.

In particular,

– The conclusions drawn by the authors depend on the relative and perhaps absolute concentration of each compound tested in their blend, and they do not have any data to indicate whether these amounts are realistic. The description of peak area as a percentage of internal standard peak area cannot provide this information without additionally calculating response factors of individual volatiles to the internal standard. (i.e., What is the ratio between [peak area internal standard/absolute amount internal standard in sample], and [peak area of volatile/absolute amount of volatile in sample], and over what range is this relationship linear; once this is established using standard curves of the volatiles of interest versus the internal standard, an equation can then be used to estimate absolute amount of volatile in sample based on the ratio of volatile peak area to internal standard peak area in each sample).

– To address this does not require absolute quantification of 50 analytes. It would be sufficient to calculate response factors for the 13 focal volatiles, for which pure standards are available. These response factors can be used to estimate amounts measured from plants and compare those to amounts tested in assays.

– It would be much easier to interpret the results of the 20 bioassays in Figure 4 if the same "low" and "high" doses had been used (not the case -- these differ by orders of magnitude), or else if these can be related to the abundance of different compounds in the measured blends, using response factors as suggested above.

2) All reviewers agree that the statistical analyses require more complete description. A statistical analysis section is missing from the methods, and tests used and replicate number n are not stated in all figure legends.

The reviewers also agree that, in several cases, the statistical analyses require revision (inappropriate tests are currently used).

Reviewer 3 provides detailed instructions. Reviewer 3 also recommends a good standard reference: Bolker et al., Trends in Ecology and Evolution, 2008.

– Figure 1: A Wilcoxon signed rank test to compare BPH settled on the two different plants is ok although more interesting approaches exist. However, Student's t-tests are not appropriate to compare number of eggs since (i) this test assumes that the dependent variable is continuous and (ii) it does not take into account the paired structure of the data. Rather use a likelihood ratio test (LR Test) or a Wald test applied on a Generalized Linear Mixed Model (GLMM, Poisson distribution error) where the tube is treated as a random factor.

– Figure 2: A chi² test is not appropriate to analyze results of olfactometer experiments (here and elsewhere). Rather use a LR Test or a Wald test applied on a GLMM (binomial distribution error) where the odor source is treated as a random factor since multiple parasitoid females were tested with the same odor source. In more detail: (subsection “Response of *A. nilaparvatae* wasps to insect-infested rice plants”): 'control plants' are compared to 'treated plants'. These are the two treatment groups. But for each group, 10 individual odor sources are used with 8-12 parasitoids tested for each odor source (= 80-120 parasitoids per experiment). The, data are structured by the treatment, but also by the individual odor source nested within the treatment. Both structure levels must be integrated in the analyses. Chi square tests compare the two treatment groups but do not account for the structuring by odor source. Not taking this into account results in pseudo-replication. The most proper way to deal with that is to include the individual odor source as a random factor in the analysis, i.e. using a mixed model. Since the outcome of the experiment is a binary response (i.e. choice for control or treated plants), a GLMM is appropriate. It should include the treatment group as a fixed factor, the odor source as a random factor, and a binomial distribution error. By the way, please clarify the protocol of the olfactometer experiment that lacks clarity (subsection “Response of *A. nilaparvatae* wasps to insect-infested rice plants).

– Figure 3: You should never conclude anything from the score plot of a PLS-DA alone, due to the fact that this analysis is prone to overfitting of the data. Any conclusion should be drawn from a dedicated significance test followed with pairwise comparisons (see e.g. Hervé, Nicolè and Lê Cao, J. Chem. Ecol. 2018). Additionally, avoid log(x+1) transformations as the constant value introduces a bias that depends on the value to which it is added (more bias for smaller values). Since zeroes occur in the data, prefer the fourth-root transformation that is not biased.

– Figures 4 and 5: Same reasoning as for Figure 2, but with filter papers rather than individual plants as odor sources.

- Figure 6: The statistical analysis is not described in the Materials and methods. It seems from the figure that Student's t-tests were used, but this is inappropriate (cf. Figure 1). Use a LR Test or Wald test applied on a GLM (Poisson distribution error) for the number of eggs, and a LR Test or Wald test applied on a GLMM (binomial distribution error) for the % of parasitized eggs (with the tube treated as a random factor).

– Figure 7: Same comment as for Figure 6, except that a GLMM should be used for both numbers of eggs and % of parasitized eggs (each time with the cage treated as a random factor).

– Figure 3—table supplement 1: Since you perform multiple testing, it is mandatory that p-values of ANOVAs are corrected (see e.g. Saccenti et al., Metabolomics 2014). I would recommend the False Discovery Rate correction.

– Figure 4—figure supplement 1: The statistical methods are not described.

In addition:

3) Please explain the addition of Shuangli Su to the author list and the re-ordering of the author list between the initial and full submission. This is fine but should be explained.

4) The field experiment was a caged choice assay using a laboratory population of parasitoids under realistic field conditions. The authors should at least comment, and if possible provide evidence or observations, regarding whether the phenomenon that SSB infestation reduces the attractiveness of rice plants to the parasitoid can be observed in the open field, given the strong learning capacity of parasitoids.

5) Reviewer 2 expressed the concern that it takes at least 5 days until the parasitized BPH eggs become red at the condition the authors set up and thus it is not clear if the parasitization rates can be reliably quantified after only 3 days. The authors should respond to this concern, if necessary with a comparison of parasitization rates calculated at 3 versus 5 days under comparable conditions.

If these five issues are addressed and the updated results can still support the authors' central claims, then we would recommend the study for publication in *eLife*. However, it is possible that addressing concerns 1 and 5 in particular will reveal that the authors' bioassays are inconclusive, and that must be determined before we can assess whether this study is suitable for publication in *eLife*.

We expect that the work required to address all five concerns can easily be completed within two months, especially if data or observations to address concern 5 are already on hand or can be cited.

[Editors' note: further revisions were suggested prior to acceptance, as described below.]

Thank you for submitting your article "Caterpillar-induced rice volatiles provide enemy-free space for the offspring of the brown planthopper" for consideration by *eLife*. Your article has been assessed by a Reviewing Editor and Ian Baldwin as the Senior Editor.

After reviewing your response to the previous round of review, the Reviewing Editor has drafted this decision to help you prepare a revised submission.

The authors have fully addressed the points raised by the prior round of peer review, but their response omits some essential information which must be provided so that the submission can be accepted for publication.

Specifically, in response to the major reviewer concern regarding their calculations, the authors state that they repeated the calculations as requested, which substantially changed their data and required new behavioral assays, but did not change their conclusions. This is fine, and the results (new values, new bioassay results) are clearly presented. However the method used to perform these new calculations is not adequately described. The authors refer to an appropriate reference (Kalambet and Kozmin, 2019; note: please correct year for this citation) which describes several acceptable methods, and do not state which of these they used, nor include their calculations in supplementary material or source data. It also cannot be determined from the new dataset how the calculations were done. This information is important both for replicability, and for understanding of interested readers regarding associated uncertainties.

The authors are thus asked to complete to their methods section and to include the new calculations, including standard curves as appropriate, as supplementary material or in the source data.

---

## [Author Response]

Essential revisions:1) The relative quantification of at least those volatiles used for bioassays needs to be done rigorously, e.g. using response factors, and the amounts of pure compounds tested in bioassays need to be compared to amounts calculated to be present in plant blends, at least based on response factors. All three reviewers agree that this is a critical point which must be resolved. Relative quantification in terms of percentage internal standard only does not support any conclusions about absolute amounts of volatiles (ng-ug emitted per plant per time) or their ratio, and thus cannot be used as a basis to select concentrations for bioassays or to interpret bioassay results.In particular,– The conclusions drawn by the authors depend on the relative and perhaps absolute concentration of each compound tested in their blend, and they do not have any data to indicate whether these amounts are realistic. The description of peak area as a percentage of internal standard peak area cannot provide this information without additionally calculating response factors of individual volatiles to the internal standard. (i.e., What is the ratio between [peak area internal standard/absolute amount internal standard in sample], and [peak area of volatile/absolute amount of volatile in sample], and over what range is this relationship linear; once this is established using standard curves of the volatiles of interest versus the internal standard, an equation can then be used to estimate absolute amount of volatile in sample based on the ratio of volatile peak area to internal standard peak area in each sample).– To address this does not require absolute quantification of 50 analytes. It would be sufficient to calculate response factors for the 13 focal volatiles, for which pure standards are available. These response factors can be used to estimate amounts measured from plants and compare those to amounts tested in assays.– It would be much easier to interpret the results of the 20 bioassays in Figure 4 if the same "low" and "high" doses had been used (not the case -- these differ by orders of magnitude), or else if these can be related to the abundance of different compounds in the measured blends, using response factors as suggested above.

This is a valid concern. In order to deal with this issue, additional experiments were conducted, in which the response factors of the 13 focal volatiles were measured. Based on the response factors, we then accurately calculated the concentrations of the 13 volatiles emitted by the plants. We indeed found that the absolute concentrations of 13 compounds calculated based on their response factors were considerably different from those detected solely based on the internal standard. The data were shown in “Figure 4—figure supplement 1”.

Based on the newly calculated concentrations of the 13 compounds, we prepared new synthetic blends for Y-tube assays with female *A. nilaparvatae* wasps. Although the concentrations of these compounds in the synthetic blends were different from the synthetic blends used previously (Figure 5—table supplement 1), the behavioral responses of the parasitoids from the new experiment (Figure 5) were quite similar to the earlier results, and the new data still fully support our conclusions.

2) All reviewers agree that the statistical analyses require more complete description. A statistical analysis section is missing from the methods, and tests used and replicate number n are not stated in all figure legends.

As explained below, most data were re-analyzed following the suggestions by reviewers. We reorganized the section “Statistical analyses”. In the revised version, all analyses are now fully described. Moreover, the tests that were used and number of replicates “n” are all mentioned in each figure legend. This should make it easier for readers to find this information.

The reviewers also agree that, in several cases, the statistical analyses require revision (inappropriate tests are currently used).Reviewer 3 provides detailed instructions. Reviewer 3 also recommends a good standard reference: Bolker et al., Trends in Ecology and Evolution, 2008.– Figure 1: A Wilcoxon signed rank test to compare BPH settled on the two different plants is ok although more interesting approaches exist. However, Student's t-tests are not appropriate to compare number of eggs since (i) this test assumes that the dependent variable is continuous and (ii) it does not take into account the paired structure of the data. Rather use a likelihood ratio test (LR Test) or a Wald test applied on a Generalized Linear Mixed Model (GLMM, Poisson distribution error) where the tube is treated as a random factor.

We agree that the LR test is more appropriate for the comparisons between number of eggs laid by BPH and the parasitism rates in our experiment. We have reanalyzed the data using LR test applied to a Generalized Linear Model (GLM, Poisson distribution error) in the current version. The outcome of these analyses provides even stronger support for our conclusions. Thank you for the suggestions.

– Figure 2: A chi² test is not appropriate to analyze results of olfactometer experiments (here and elsewhere). Rather use a LR Test or a Wald test applied on a GLMM (binomial distribution error) where the odor source is treated as a random factor since multiple parasitoid females were tested with the same odor source. In more detail: (subsection “Response of A. nilaparvatae wasps to insect-infested rice plants”): 'control plants' are compared to 'treated plants'. These are the two treatment groups. But for each group, 10 individual odor sources are used with 8-12 parasitoids tested for each odor source (= 80-120 parasitoids per experiment). The, data are structured by the treatment, but also by the individual odor source nested within the treatment. Both structure levels must be integrated in the analyses. Chi square tests compare the two treatment groups but do not account for the structuring by odor source. Not taking this into account results in pseudo-replication. The most proper way to deal with that is to include the individual odor source as a random factor in the analysis, i.e. using a mixed model. Since the outcome of the experiment is a binary response (i.e. choice for control or treated plants), a GLMM is appropriate. It should include the treatment group as a fixed factor, the odor source as a random factor, and a binomial distribution error. By the way, please clarify the protocol of the olfactometer experiment that lacks clarity (subsection “Response of A. nilaparvatae wasps to insect-infested rice plants).

We agree with the reviewer. The data were reanalyzed by a LR test applied to a GLM in the current version. As suggested, the “protocol of the olfactometer experiment” was further clarified (subsection “Response of *A. nilaparvatae* wasps to insect-infested rice plants”).

– Figure 3: You should never conclude anything from the score plot of a PLS-DA alone, due to the fact that this analysis is prone to overfitting of the data. Any conclusion should be drawn from a dedicated significance test followed with pairwise comparisons (see e.g. Hervé, Nicolè and Lê Cao, J. Chem. Ecol. 2018). Additionally, avoid log(x+1) transformations as the constant value introduces a bias that depends on the value to which it is added (more bias for smaller values). Since zeroes occur in the data, prefer the fourth-root transformation that is not biased.

We agree that we cannot draw conclusions based on the score plot of a PLS-DA alone. In fact, with the PLS-DA analysis, our objective is simply to provide an overview of the differences in the volatile compounds among the treatments. Figure 3 shows that the PLS-DA model can indeed separate the volatile profiles released by differently treated plants.

Further details regarding volatile profiles are shown in Figure 3—table supplement 1, in which the significance tests (one-way ANOVA) were conducted followed with pairwise comparisons as suggested by the reviewer for the data of relative volatile quantities. We also agree with the reviewer to use fourth-root transformation instead of log(x+1) before analysis by one-way ANOVA. And the results of this are updated in the supplementary table.

In addition, according to *eLife* policy, any supplementary table or figure has to be linked to a table or figure in main text. Since we do not think it is good to include the table regarding volatile contents in the main text, we decided to leave the PLS-DA figure in the main text and the table is linked to the figure as a supplementary table.

– Figures 4 and 5: Same reasoning as for Figure 2, but with filter papers rather than individual plants as odor sources.

The data were reanalyzed using LR tests applied to a GLM as suggested. Thanks for the suggestion.

- Figure 6: The statistical analysis is not described in the Materials and methods. It seems from the figure that Student's t-tests were used, but this is inappropriate (cf. Figure 1). Use a LR Test or Wald test applied on a GLM (Poisson distribution error) for the number of eggs, and a LR Test or Wald test applied on a GLMM (binomial distribution error) for the % of parasitized eggs (with the tube treated as a random factor).

We agree and LR tests applied to a GLM were used to compare the number of eggs and the percentage of parasitized eggs.

– Figure 7: Same comment as for Figure 6, except that a GLMM should be used for both numbers of eggs and % of parasitized eggs (each time with the cage treated as a random factor).

GLMs were used to compare the number of eggs and the percentage of parasitized eggs as suggested.

– Figure 3—table supplement 1: Since you perform multiple testing, it is mandatory that p-values of ANOVAs are corrected (see e.g. Saccenti et al., Metabolomics 2014). I would recommend the False Discovery Rate correction.

The data were re-analyzed using one-way ANOVA followed by Tukey HSD test, and, as suggested, the p-values were corrected according to the False Discovery Rate correction.

– Figure 4—figure supplement 1: The statistical methods are not described.

The statistical method has been added.

In addition:3) Please explain the addition of Shuangli Su to the author list and the re-ordering of the author list between the initial and full submission. This is fine but should be explained.

The experiments were mainly conducted by the first author Xiaoyun Hu. While Shuangli Su (a master student) helped Xiaoyun Hu to do some experiments regarding parasitoid responses to plant volatile compounds (Figure 4). In the initial submission, she was left out by mistake. She was therefore added in the full submission as the second co-author.

4) The field experiment was a caged choice assay using a laboratory population of parasitoids under realistic field conditions. The authors should at least comment, and if possible provide evidence or observations, regarding whether the phenomenon that SSB infestation reduces the attractiveness of rice plants to the parasitoid can be observed in the open field, given the strong learning capacity of parasitoids.

Good point.

As for the parasitoids, although they were from a laboratory colony, field collected individuals are introduced yearly. We therefore believe that they still exhibit their natural responses. Also, although we did not investigate parasitoid responses in the fields in the current study, we can indirectly deduce from previous papers that SSB infestation reduces the attractiveness of rice plants to the parasitoid also under field condition. Recent studies on risk assessment of GM rice show higher densities of planthoppers on non-Bt rice plants due to higher SSB damage (Wang et al., 2018). Based on these results, it could be expected that the planthopper parasitoid *A. nilaparvatae* will be more abundant on non-Bt rice with higher densities of SSB and planthoppers. Yet, field studies show that parasitoids densities are similar for Bt and non-Bt rice fields (Tian et al., 2008. J Environ Entomol). We can speculate that this is due to the reduced attractiveness of rice plants when SSB is present.

Moreover, specialist parasitoids have much poorer learning capacities and a shorter-lived memory compared to generalist parasitoids, and rely mostly on fixed responses to host associated cues (Vet and Dicke, 1992; Annu Rev Entomol; Guo and Wang, 2019). We can therefore expect that the specialist parasitoid *A. nilaparvatae* is also not very flexible in its responses. We now address this in the Discussion (third paragraph).

5) Reviewer 2 expressed the concern that it takes at least 5 days until the parasitized BPH eggs become red at the condition the authors set up and thus it is not clear if the parasitization rates can be reliably quantified after only 3 days. The authors should respond to this concern, if necessary with a comparison of parasitization rates calculated at 3 versus 5 days under comparable conditions.

This is a valid concern. This misunderstanding is due an unclear description of the experiment, which is entirely our fault. In fact, the rice plants with planthopper eggs were exposed to the parasitoids for 72h and two days later, the total number of BPH eggs on each plant was counted and their parasitization status was determined under a microscope. In the revised text, the relevant sentences have been revised (subsection “Parasitism rates of *N. lugens* eggs by *A. nilaparvatae* wasps”).

[Editors' note: further revisions were suggested prior to acceptance, as described below.]

The authors have fully addressed the points raised by the prior round of peer review, but their response omits some essential information which must be provided so that the submission can be accepted for publication.Specifically, in response to the major reviewer concern regarding their calculations, the authors state that they repeated the calculations as requested, which substantially changed their data and required new behavioral assays, but did not change their conclusions. This is fine, and the results (new values, new bioassay results) are clearly presented. However the method used to perform these new calculations is not adequately described. The authors refer to an appropriate reference (Kalambet and Kozmin, 2019; note: please correct year for this citation) which describes several acceptable methods, and do not state which of these they used, nor include their calculations in supplementary material or source data. It also cannot be determined from the new dataset how the calculations were done. This information is important both for replicability, and for understanding of interested readers regarding associated uncertainties.The authors are thus asked to complete to their Materials and methods section and to include the new calculations, including standard curves as appropriate, as supplementary material or in the source data.

We agree that we were not very precise in describing the calculations of volatile concentrations. We tried to keep text short and therefore just cited the paper of Kalambet and Kozmin, 2018. Although the paper does describe the method that we used, we agree that it is not sufficiently clear for readers to know which method we exactly used. In the new version of our paper, we provided the requested details in the Materials and methods section and we added a new reference that very clearly describes the method that we used. In addition, we added the data and standard curves (plots) that were used to determine the “response factors” of the 13 relevant compounds (see file Figure 5—figure supplement 1—source data 1). We believe that this additional information will guarantee the replicability and the understanding of the methodology by the readers.

As for the year of the cited paper Kalambet and Kozmin, 2018, we rechecked the paper and found the paper was indeed published in 2018, and it was correctly cited.